# Photon-Detection-Probability Simulation Method for CMOS Single-Photon Avalanche Diodes

**DOI:** 10.3390/s20020436

**Published:** 2020-01-13

**Authors:** Chin-An Hsieh, Chia-Ming Tsai, Bing-Yue Tsui, Bo-Jen Hsiao, Sheng-Di Lin

**Affiliations:** Institute of Electronics, National Chiao Tung University, Hsinchu 30010, Taiwan; eric50224@gmail.com (C.-A.H.); cmtsai@mail.nctu.edu.tw (C.-M.T.); bytsui@mail.nctu.edu.tw (B.-Y.T.); y0718@nctu.edu.tw (B.-J.H.)

**Keywords:** single-photon avalanche diode (SPAD), photon-detection probability, CMOS technology

## Abstract

Single-photon avalanche diodes (SPADs) in complementary metal-oxide-semiconductor (CMOS) technology have excellent timing resolution and are capable to detect single photons. The most important indicator for its sensitivity, photon-detection probability (PDP), defines the probability of a successful detection for a single incident photon. To optimize PDP is a cost- and time-consuming task due to the complicated and expensive CMOS process. In this work, we have developed a simulation procedure to predict the PDP without any fitting parameter. With the given process parameters, our method combines the process, the electrical, and the optical simulations in commercially available software and the calculation of breakdown trigger probability. The simulation results have been compared with the experimental data conducted in an 800-nm CMOS technology and obtained a good consistence at the wavelength longer than 600 nm. The possible reasons for the disagreement at the short wavelength have been discussed. Our work provides an effective way to optimize the PDP of a SPAD prior to its fabrication.

## 1. Introduction

Since the birth in the 1960s, single-photon avalanche diodes (SPADs) advanced with Si-based semiconductor technology [1,2]. Driven by fast-developing CMOS technology, detection of extremely weak light using SPADs has been a growing field in the past two decades [3,4,5,6]. Due to their single-photon sensitivity and excellent timing resolution, SPADs have been used in areas such as fluorescence lifetime imaging microscopy [7,8,9,10], light detection and ranging (LiDAR) [11,12], radiometric temperature measurement [13], and time-gated Raman spectroscopy [14]. Very recently, the strong demand for LiDAR for autonomous driving (AD) or advanced driver assistance system (ADAS) requires high-resolution 3-D images on a targets at distance up to 100–200 m to ensure car safety. Long-distance ranging within a short time interval (<0.1 ms) acquires high-power pulsed lasers (peak power > 100 W) that not only increases laser module cost and complexity but also causes the concern on eye safety issue. SPADs with good photon-detection probability (PDP) is highly desirable for this application as their single-photon sensitivity can dramatically reduce needed laser power. However, to optimize the PDP in a wavelength range aiming for certain applications, such as 905 nm for Si-based LiDAR, it needs a reliable tool based on the fundamental device physics and the material parameters. In fact, although there were a few theoretical models for the PDP calculation [15,16,17], a quantitative and reliable prediction of PDP has been a challenging task due to its complicated factors, including the doping profiles, the carrier transport above breakdown voltage, the non-uniform electric-field and impact-ionization distributions, and the spatially-dependent breakdown triggering probability. Pancheri et al. compared two breakdown models to simulate bias-dependent PDP and an excellent consistence between their calculation and experiment was achieved [17] with a few fitting parameters. The free-running fitting parameters, however, make a quantitative PDP prediction not possible in device optimization. Here, we propose and demonstrate a parameter-free simulation method to predict PDP of SPADs based only on CMOS fabrication parameters. By combining conventional optical/electrical TCAD (technology computer-aided design) with position-dependent breakdown-triggering probability calculation, the voltage-dependent PDP spectra have been simulated. We have compared our simulation with the experimental results in a customized 800-nm CMOS technology and obtained a good consistence. Our work provides a feasible method to engineer and to optimize PDP of CMOS SPADs which is crucial for enhancing the performance of these intriguing devices.

## 2. Methods and Simulation Results

A procedure combining the conventional TCAD (Synopsys Sentaurus) [18] with the 1-D triggering probability calculation [19] to simulate PDP has been developed. It includes the following steps: (1) the doping profile simulation using SPROCESS module based on the process parameters and the device layouts, (2) the I-V and electric field distributions at various voltages using SDEVICE module, (3) the 1-D triggering probability calculation using MATLAB with the obtained electric field distributions, (4) the photo-generation simulation at the certain wavelengths at the bias voltage without avalanche gain using SDEVICE module, and (5) the PDP spectra at various excess voltage Vex can be obtained accordingly. For clarity, we shall present our simulation results along with the experimental counter parts in details as follows.

### 2.1. Device Structure, Doping Proflies, and Electric Field Distribution

Our customized SPADs in circular shape with a diameter of 20 µm were fabricated in the Episil 800-nm CMOS technology. Figure 1a illustrates the schematic cross-sectional device structure. The P-N junction is formed by the Pabs and n-type buried layer (NBL) layers with the Pwell layer as a guard ring. In fact, the Pabs (‘abs’ for absorption) layer is the only customized layer in this design. With the provided process parameters, we performed process simulation and obtained the 2-D doping profiles as shown in Figure 1b. The heavily p-type doped region locates at the P+ region and the upper part of Pabs layer. A clearer picture can be seen from the cross-sectional doping profile in z-direction in Figure 1c. The doping profile at the center of device (*r* = 0 µm) exhibits a double-peak distribution of the Pabs layer, coming from the two ion implantations, BF_23_ (5 × 10^14^ cm^−2^, 40 keV) and Boron (3 × 10^13^ cm^−2^, 850 keV), applied. The shallow one was used to compensate the existing Nwell, making it p-typed near the surface. The deep one formed the P-N junction at the depth of about 2.4 µm with the underneath n-type buried layer (NBL) layer. Figure 1c also shows the doping profile at the device edge (*r* = 10 µm, dashed line). A very small difference near the junction is spotted but this difference and the edge effect will make a substantial drop of the resultant PDP discussed later. It is worth noting that the doping profiles have been confirmed with the secondary ion mass spectrometer (SIMS) and a very good consistence was revealed (not shown here).

We used SDEVICE module for the bias-dependent electric field (E-field) and current-voltage (I-V) characteristic simulations. Figure 2a shows the 2-D distribution of E-field strength, $\left|\vec{E}\right|$, of the device biased at 28.5 V. As expected, the strongest E-field locates at the Pabs/NBL junction. Figure 2b exhibits the 1-D E-field strength profile as a function of the depth. The maximum E-field strength at the device center (*r* = 0 µm) reaches 415 kV/cm at the depth of about 2.4 µm. The depletion region ranges from ~1.9 to ~2.9 µm so the depletion width of about 1 µm is equally shared by the Pabs and NBL layers. Plotted in dashed line in Figure 2b, the maximum E-field strength at the device edge (*r* = 10 µm) is only ~394 kV/cm at the same depth. The weaker E-field at the edge arises from the slight difference of doping profile and the guard ring effect. More importantly, this weakened E-field significantly lowers the impact ionization as well as the device PDP because of the strong dependence between E-field strength and impact ionization coefficient. In the inset on Figure 2b, we plotted the simulated current-voltage (I-V) curve. An abrupt breakdown occurs at the breakdown voltage (Vbd) of ~28.5 V. Note that, above breakdown voltage, the E-field distributions have also been obtained by turning off the impact ionization effect in the SDEVICE module.

### 2.2. Breakdown Trigger Probability

When an electron-hole pair is generated by an incident photon, both of them could trigger a breakdown and deliver an event signal to the readout circuit. This breakdown trigger probability depends on the location of photo-generation and the electric field distribution in the device. With the E-field distributions at all excess voltages Vex, the bias- and depth-dependent 1-D trigger probability in the depletion region can be solved at all *r*-positions. In this work, we used Okuto and Crowell model for impact ionization coefficients [18,19]. The local-field model was applied to calculate the breakdown trigger probability of electrons ($P_{e}\left(z\right)$) and holes ($P_{h}\left(z\right)$) by solving the following equations [20].
(1)$$\left\{\begin{array}{l} \frac{dP_{e}}{\mathrm{dz}}=-\left(1-P_{e}\right)\alpha_{e}\left[P_{e}+P_{h}-P_{e}P_{h}\right]\\ \frac{dP_{h}}{\mathrm{dz}}=\left(1-P_{h}\right)\alpha_{h}\left[P_{e}+P_{h}-P_{e}P_{h}\right] \end{array}\right.$$

The parameters *α_e_* and *α_h_* are respective impact ionization coefficients of electron and hole, both dependent on the local electric field. The used boundary conditions are $P_{e}\left(z_{n}\right)=0$ and $P_{h}\left(z_{p}\right)=0$, where *z*_n_ and *z*_p_ are the locations of the depletion region edges on the n- and p-side, respectively. A simple MATLAB code using the existing solver BVP4C was prepared to get a self-consistent solution of Equation (1) based on the previously obtained Vex- and *r*-dependent E-field distributions. With the solutions $P_{e}\left(z\right)$ and $P_{h}\left(z\right)$, we calculated the *z*-dependent total trigger probability $P_{t}\left(z\right)$ simply by,
(2)$$P_{t}=P_{e}+P_{h}-P_{e}P_{h}.$$

This procedure was repeated to get the trigger probability at all excess voltages Vex (0–5 V) and at various *r*-positions (*r* = 0, 4, 6, 7, 8, 9, and 10 µm).

The resultant electron, hole, and total trigger probability profiles at *r* = 0 µm at Vex = 1.0, 3.0, and 5.0 V are shown in Figure 3a–c, respectively. We can see that all *P_e_* and *P_h_* meet the boundary condition of $P_{e}\left(z_{n}\right)=0$ and $P_{h}\left(z_{p}\right)=0$ as expected. The highest electron trigger probability is larger than that of hole’s due to the larger impact ionization coefficient of electron. Figure 3a shows that, at Vex =1.0 V, the total trigger probability P_t_ is about 0.25 at the p-side depletion edge (depth ~1.93 µm) and drops to about 0.05 at the n-side depletion edge (depth ~2.86 µm). With the increasing excess voltage Vex, all trigger probabilities increase substantially. The maximum P_t_ goes up to about 0.73 at Vex = 3.0 V in Figure 3b and to about 0.9 at Vex = 5.0 V in Figure 3c, all occurring at the p-side depletion edge. The high trigger probability at a moderate excess voltage has been achieved with our design and it is crucial for enhancing the PDP.

Figure 4a–c illustrates the depth-dependent total trigger probability (*P_t_*) at the three excess voltages. At Vex = 1.0 V in Figure 4a, the trigger probabilities decrease with the increasing *r*-positions. Particularly, a dramatic drop to nearly zero at *r* = 10 µm is seen. As the Vex increases, the trigger probabilities at *r* = 10 µm increases significantly. At Vex = 5.0 V in Figure 4c, the trigger probabilities at *r* = 10 µm are lower than others. This result indicates that, for our SPAD, the non-uniformity of PDP caused by the edge effect does exist but will be eased at high excess voltage. This non-uniformity will be considered in our simulation later.

### 2.3. Anti-Reflection Coating and Photo-Generation Rate Distribution

The PDP of a P-N junction SPAD can be calculated by [16,17],
(3)$$\mathrm{PDP}\;\left(\lambda \right)=\left(1-R\left(\lambda \right)\right)\int_{Zp}^{Zn}\alpha \left(\lambda \right)e^{-\alpha \left(\lambda \right)z}P_{t}\left(z\right)dz+\eta_{ph}^{e}\left(\lambda ,z_{p}\right)P_{e}\left(z_{p}\right)+\eta_{ph}^{h}\left(\lambda ,z_{n}\right)P_{h}\left(z_{n}\right).$$

There are three terms in this equation, respectively corresponding to three contributed sources of PDP: (1) the electron-hole pairs generated in the depletion region, (2) the electrons generated in the p-type neutral region, and (3) the holes generated in the n-type neutral region. In Equation (3), $R\left(\lambda \right)$ is the wavelength-dependent surface reflectivity, α(λ) is the absorption coefficient of silicon at wavelength of λ, *z_n_* and *z_p_* are the depth of respective depletion edges on the n- and p-side, $P_{t}\left(z\right)$ is the total breakdown probability triggered by electron-hole pairs photo-generated at *z* position. $P_{e}\left(z_{p}\right)$ and $P_{h}\left(z_{n}\right)$ are the electron and hole breakdown probabilities triggered by electrons and holes at *z_p_* and *z_n_*, respectively. $\eta_{ph}^{e}\left(\lambda ,z_{p}\right)$ is the number of electrons that are generated in p-type neutral region by one photon with wavelength λ and then diffuses to the p-side depletion edge (*z_p_*), called electron quantum efficiency for p-type neutral region, so they can trigger a breakdown event from there with the trigger probability $P_{e}\left(z_{p}\right)$. Likewise, $\eta_{ph}^{h}\left(\lambda ,z_{n}\right)$ is the number of holes that are generated in n-type neutral region by one photon with wavelength λ and then diffuses to the n-side depletion edge (*z_n_*), called hole quantum efficiency for n-type neutral region, so they can trigger a breakdown event from there with the trigger probability $P_{h}\left(z_{n}\right)$. These two numbers, defined by us, are of course less than 1 and their values are extracted with the TCAD tool as detailed in the following.

First, on the surface of our SPADs, we designed and fabricated an anti-reflection coating (ARC) layer. The single-layer ARC used SiNx (thickness ~61 nm) and the aiming wavelength was 500 nm. Figure 5 shows the simulated and measured reflectivity spectra. The reflectivity measurement was performed on a 2 mm × 2 mm area specified for ARC layer on the same wafer by using a commercial equipment (N&K 1500). A good consistence between the simulation and the experiment is seen. The reflectivity dips at 504 nm with a value less than 0.1% and rises to about 16.5% at 900 nm.

To estimate $\eta_{ph}^{e}$ and $\eta_{ph}^{h}$, we simulated the photo-generation current with the ARC layer on the top of our SPAD by using the SDEVICE module. For each wavelength, the device was illuminated by the photons at a constant power density of *P*_0_ = 0.05 W/cm^2^. With the steady-state solution, we obtained the photo-generation current distributions at all wavelengths. It is worth noting that we found that the distributions are nearly unchanged with the biased voltages of 0, 10, and 20 V in our device. In other words, the voltage dependence of PDP spectra in Equation (3) solely comes from the trigger probability function. The steady-state current density distributions of the device biased at 20 V are shown in Figure 6. Figure 6a–c exhibits the 2-D total, electron, and hole current density distributions under 700-nm illumination, respectively. The current density is lower at the center at the depth away from the depletion region indicated by the white lines. This is caused by the side-located electrical contacts but no effect on our simulation because the electron and hole currents were taken at the edge of depletion region. For clarity, in Figure 6d, we plot their cross-sectional 1-D profiles in depth (*z*-direction) at device center (*r* = 0 µm). The depletion region is also plotted in this figure. The electron current density diffuses from the upper p-typed neutral region is the value at the p-side depletion edge (*z_p_*) at ~1.94 µm, as pointed by the arrow in red. Likely, the hole current density diffuses from the upper n-typed neutral region is the value at the n-side depletion edge (*z_n_*) at ~2.86 µm, as pointed by the arrow in blue. Taking the respective electron and hole current density at the depletion edge, the values of $\eta_{ph}^{e}$ and $\eta_{ph}^{h}$ were calculated accordingly. Figure 6e,f shows the obtained result as a function of *r*-position at the incident wavelengths of 500, 700, and 900 nm. In Figure 6e, the electron quantum efficiency $\eta_{ph}^{e}$ for 500-nm photons at the center is as high as 0.94. This value is so large because of the absorption coefficient of Si at 500 nm is ~1.11 × 10^4^/cm that gives the absorption depth of ~0.9 µm. Therefore, the most 500-nm photons are absorbed in the p-type region that extends to the depth of ~1.9 µm. For the same reason, the hole quantum efficiency $\eta_{ph}^{h}$ for 500-nm photons in Figure 6f is as low as 0.003 as very few photons can go deep enough to reach the n-type neutral region (>2.9 µm). However, one has to be cautious that the collection efficiency in the neutral region in TCAD simulation could have been overestimated due to that the defect density generated in process simulation could be much lower than that in practice. Caused by the previously mentioned edge effect, the electron or hole quantum efficiency is near constant across the device for *r* < 9 µm but it drops quickly at the device edge. For the 700-nm and 900-nm photons, the respective absorption coefficients are 7.9 × 10^3^/cm and 3.06 × 10^2^/cm, and the corresponding absorption depths are 5.26 and 32.7 µm. As shown in Figure 6e,f, for the 700-nm photons, the center (*r* = 0 µm) electron quantum efficiency is ~0.31 and the hole’s is ~0.082. For the 900-nm photons, the center (*r* = 0 µm) electron quantum efficiency is ~0.045 and the hole’s is ~0.021. It is interesting to spot that the hole quantum efficiency for the 900-nm photons is higher than that for 500-nm ones, which is attributed to the larger absorption depth at 900 nm.

With obtained electron and hole quantum efficiencies $\eta_{ph}^{e}$ and $\eta_{ph}^{h}$ at all wavelengths, the reflectivity spectra in Figure 5, and the trigger probability functions previously simulated, the PDP spectra at a fixed voltage can be calculated by using Equation (3) at last. To simulate bias-dependent PDP, we repeated the calculation of trigger probability functions based the electric field distributions at all excess voltages Vex and used the same $\eta_{ph}^{e}$ and $\eta_{ph}^{h}$ and reflectivity spectra. For clarity, the simulated PDP will be presented together with the measured ones in the following experiment section.

## 3. Experimental Results and Discussions

Our SPADs were fabricated on a 6-inch wafer in the Episil 800-nm 40-V CMOS technology. Aside from our devices and their accompanying circuits, the areas used for checking the reflectivity spectra and the doping concentration profiles were also placed on the same wafer. The characteristics given here were taken from the most common device.

### 3.1. Experimental Method and Results

With an NMOS (n-channel metal-oxide-semiconductor) passive quenching circuit on chip, the anode of our SPAD was connected to an inverter and a buffer for signal output. Figure 7 shows the measured dark-count rate (DCR) as a function of the bias voltage. The breakdown voltage was about 29 V and the DCR was in the range of a few tenths kHz and exhibited a linear dependence on the excess voltage Vex. The DCR is a bit high as the process parameters of the customized layer has not been optimized. To characterize the PDP in a dark box, photons from a halogen light source (ORIEL-66188) were first dispersed by a monochromator (HR-550) and then coupled into a large-core bifurcated fiber (MWG-100S-SD). The light output from the two fiber ends was respectively guided to two quartz tube with a square cross section (2.8 mm × 2.8 mm) of and a length of 40 mm to get the normally-incident illumination. Two 2× lenses (MML2-ST40) were placed between the quartz tube and the SPADs and the calibration photodiode (PD, Thorlabs FDS100), respectively, to enlarge the uniform illumination area to about 5.6 mm × 5.6 mm. The SPAD and the PD were illuminated at the same time so the incident photon flux was real-time monitored by the PD for calculating the incident photon flux per unit area on the SPAD. A shutter in front of the monochromator was switched on and off in turns so the light and dark counts were taken under the same condition.

### 3.2. Comparison between Experimental and Thoeretical PDP

Figure 8 shows the measured PDP spectra in 400–900 nm, together with the simulated ones at the excess voltages Vex = 1.0, 3.0, and 5.0 V. Let us focus on the experimental data first. The peak PDP occurs at about 575 nm and the respective values are 36%, 53%, and 58% at Vex = 1.0, 3.0, and 5.0 V. These values are comparable to the best ones previously reported [21,22]. The single-layer ARC gives a smooth rather than oscillating spectra, which is good for using this SPAD at various wavelength ranges as the PDP is more stable. The PDP at 900 nm is about 9% at Vex = 5.0 V, which is applicable for LiDAR but could be enhanced simply by changing the thickness of the ARC layer.

Comparing the experimental and simulated results, we can see a good consistence in the wavelength range larger than 600 nm. However, a clear difference exists in the short wavelength range. The simulated PDPs reach their peak values at about 500 nm and the highest one is about 80% at Vex = 5.0 V. We think there are two possible reasons for this discrepancy between the experiment and simulation. First one is that the underestimated surface recombination velocity at the silicon/SiNx interface. The interface was etched before the ARC layer deposition so the surface damage could be much more severe. The other reason is the overestimated diffusion length in the p-typed neutral region. The Pabs layer was customized for our SPAD so it has not been optimized to minimize its defect density. This also explains the higher DCR observed. The minority carrier lifetime in the Pabs layer could be much shorter, as well as its diffusion length. At short wavelength (<600 nm), the PDP is mostly contributed by the diffusive electron from the Pabs layer so the experimental PDP decreased. In this work, we simply used the default values in TCAD for the mentioned parameters. Further works on tuning those parameters are being undertaking.

We have also examined the bias-dependent PDP. Figure 9a shows the experimental and simulated bias-dependent PDP at 500, 700, and 900 nm. Again, there is a clear difference between the 500-nm ones as seen in Figure 8. The data for 700 nm and 900 nm show a better consistence. To see the trend more clearly, we have normalized the experimental PDP at each wavelength with a scaling factor to make the experimental and simulated PDP at Vex ~5.0 V equal, as shown in Figure 9b. The PDP at 900 nm exhibits an excellent fit but the 500 nm and 700 nm ones do not. Further investigations are needed to clarify this discrepancy.

### 3.3. PDP Pie-Chart Analysis

To further understand the components of PDP, the pie-chart analysis on our device based on the simulation result was performed. In Figure 10, we only plot the pie-charts for the 700 and 900 nm PDP analysis at Vex = 5.0 V considering the substantial discrepancy between the experiment and simulation at 500 nm. The simulated PDPs at Vex = 5.0 V are 35.4% at 700 nm and 6.0% at 900 nm. Figure 10a exhibits the 700 nm PDP contributions from the photo-carriers generated in the depletion region (black), in the upper Pabs neutral region (red), and in the lower NBL neutral region (green). Clearly, the most PDP comes from the P-neutral region absorption (~78%). The depletion region contributes about 16% and the N-neutral region gives about 6%. At 900 nm, as shown in Figure 10b, because of the deeper absorption depth, the contribution of the depletion and the N-neutral region increase but the P-neutral region still dominates due to the high trigger probability (~0.9) at the p-side depletion edge (Figure 3c). On the other hand, in order to improve PDP at a certain wavelength, one needs to identify the loss mechanism of photon detection. We divided the loss into five categories, photons reflected by the surface (reflection), photons go through the P-N junction and beyond the bottom NBL layer without absorption (transmission), and photons absorbed in the three regions (depletion, P-neutral, and N-neutral) but without triggering a breakdown event. Figure 10c,d shows the percentage of each category at 700 nm and at 900 nm, respectively. The main loss at both wavelengths is caused by transmission. In typical CMOS technology, all layers are not deeper than 5 µm so the absorption for near infrared photons is not enough. This issue becomes much worse at 900 nm as nearly 66% photons simply pass the bottom NBL layer, which seriously limits its PDP. A good PDP at 905 nm is crucial for low-cost vehicle LiDAR application so a deeper absorption layer could be a way out of this problem.

## 4. Conclusions

We have presented a parameter-free method for PDP simulation for CMOS SPADs. Combining the process, electrical, and optical simulations in commercial TCAD software and the calculation of the electric-field-dependent breakdown trigger probability, the bias-dependent PDP spectra have been simulated. The possible causes of the discrepancy between the simulated and experimental data have been discussed. Our method provides a way to design and to optimize the PDP of CMOS SPADs.

## Figures and Tables

**Figure 1 sensors-20-00436-f001:**
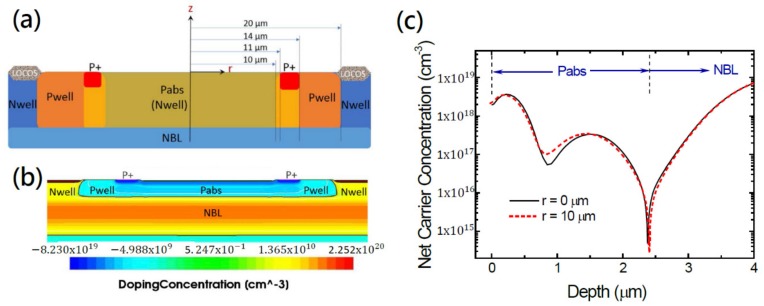
(**a**) Schematic cross-sectional device structure. The SPAD active region is 20 µm in diameter. The distance from the device center is denoted as *r*. Note that there is a 4-µm overlap region between the Pabs and Pwell layers. (**b**) Simulated 2-D doping profile. (**c**) *z*-direction 1-D net doping concentration profiles taken at *r* = 0 µm (black solid line) and *r* = 10 µm (red dashed line).

**Figure 2 sensors-20-00436-f002:**
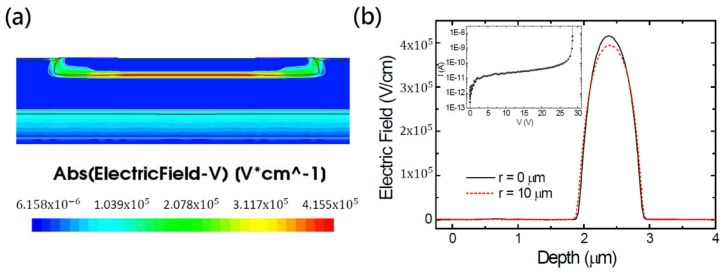
(**a**) Simulated 2-D electric-field strength distribution of the single-photon avalanche diodes (SPAD) biased at 28.0 V; (**b**) Cross-sectional electric-field strength profiles at device center (*r* = 0 µm, solid black line) and at the edge (*r* = 10 µm, red dashed line), respectively, as a function of the depth. Inset on Figure 2b: simulated I-V curve.

**Figure 3 sensors-20-00436-f003:**
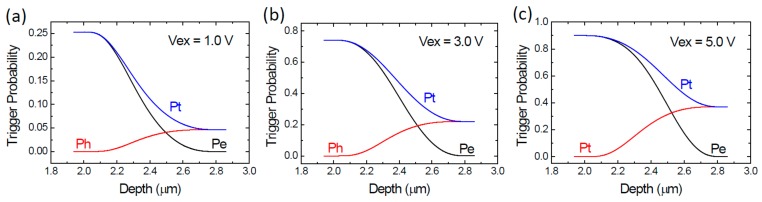
Simulated electron (*P_e_*), hole (*P_h_*), and total (*P_t_*) trigger probabilities as a function of the depth at device center (*r* = 0 µm) at the three excess voltages Vex. (**a**) Vex = 1.0 V; (**b**) Vex = 3.0 V; (**c**) Vex = 5.0 V.

**Figure 4 sensors-20-00436-f004:**
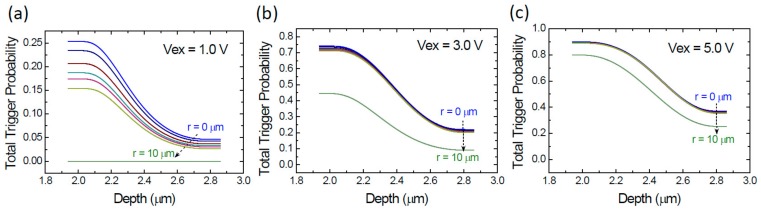
Simulated total trigger probability (*P_t_*) as a function of the depth at various *r*-positions including *r* = 0, 4, 6, 7, 8, 9, and 10 µm and at the three excess voltages Vex. (**a**) Vex = 1.0 V; (**b**) Vex = 3.0 V; (**c**) Vex = 5.0 V.

**Figure 5 sensors-20-00436-f005:**
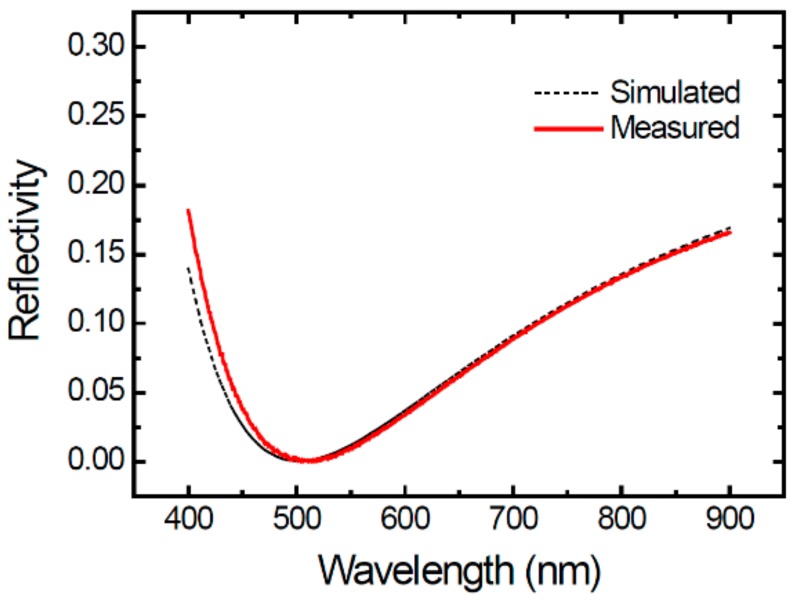
Simulated and measured normal-incidence reflectivity as a function of wavelength.

**Figure 6 sensors-20-00436-f006:**
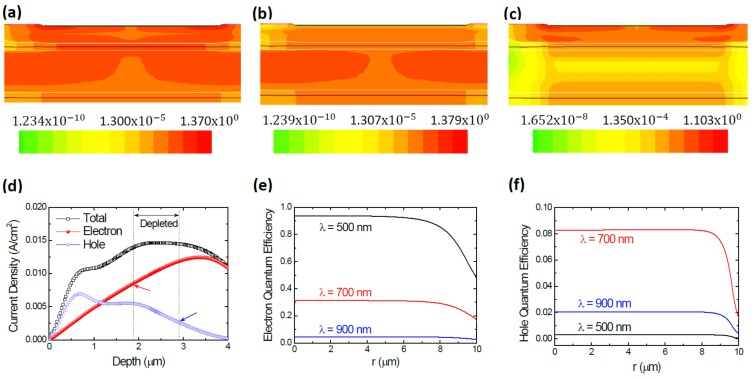
Simulated steady-state current distributions of the device under illumination and the corresponding electron and hole quantum efficiencies. (**a**) 2-D total current density distribution under the 700-nm illumination; (**b**) 2-D electron current density distribution under the 700-nm illumination; (**c**) 2-D hole current density distribution under the 700-nm illumination; (**d**) Cross-sectional 1-D profiles of total, electron, and hole current densities as a function of the depth *z* at device center under the 700-nm illumination; (**e**) Electron quantum efficiency as a function of the *r*-position under the 500-, 700-, and 900-nm illuminations; (**f**) Hole quantum efficiency as a function of the *r*-position under the 500-, 700-, and 900-nm illuminations. The unit of color bars in (**a**–**c**) is A/cm^2^.

**Figure 7 sensors-20-00436-f007:**
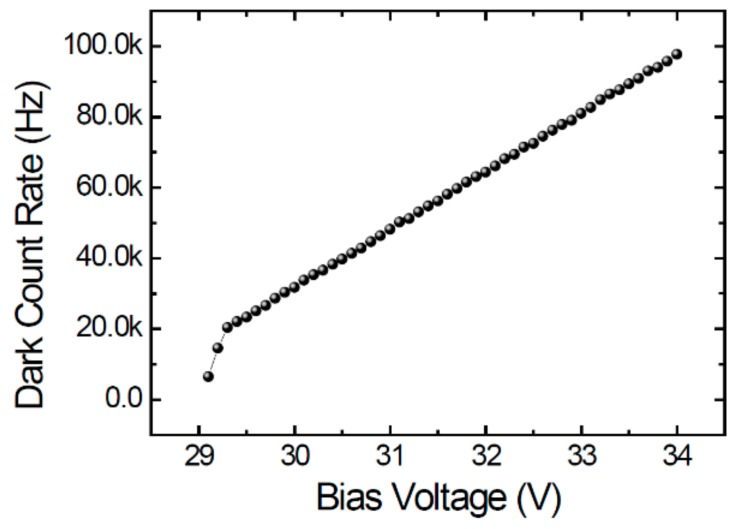
Measured dark-count rate (DCR) of the SPAD as a function of bias voltage.

**Figure 8 sensors-20-00436-f008:**
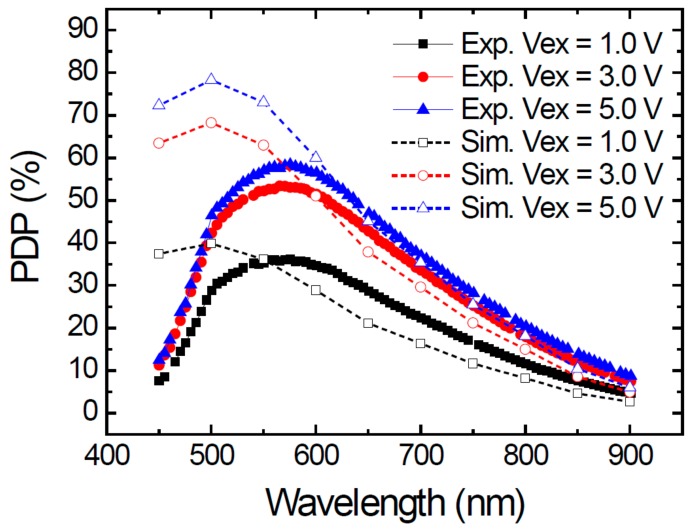
Experimental and simulated photon-detection probability (PDP) spectra at excess voltages Vex = 1.0, 3.0, and 5.0 V.

**Figure 9 sensors-20-00436-f009:**
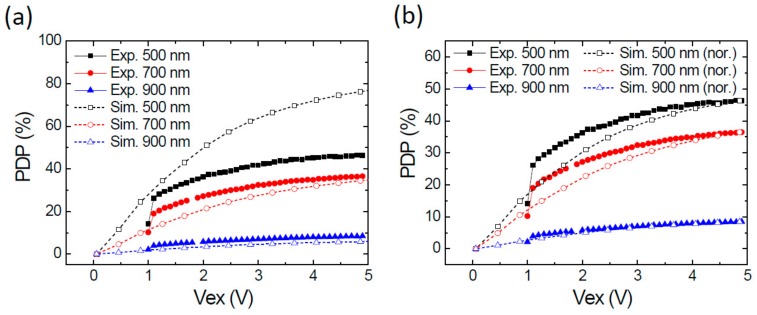
Bias-dependent PDP at the wavelengths of 500, 700, and 900 nm. (**a**) Measured and simulated PDP; (**b**) Original measured PDP and simulated PDP normalized with the corresponding PDP at Vex = 5.0 V.

**Figure 10 sensors-20-00436-f010:**
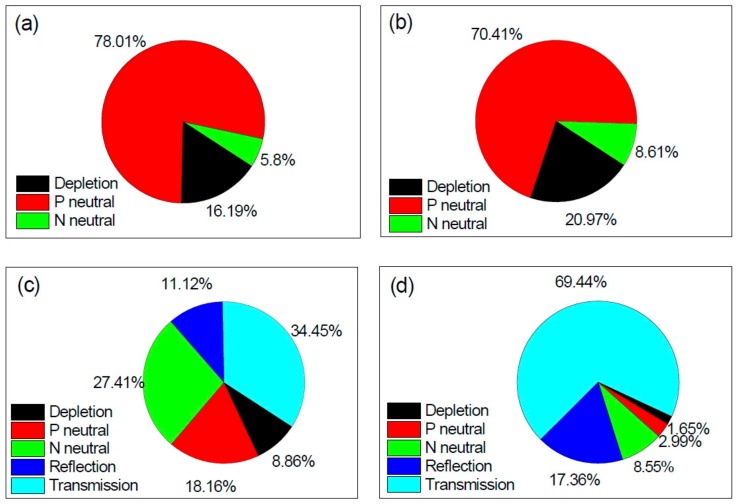
Simulated pie-charts of PDP components of our SPAD biased at Vex = 5.0 V. (**a**) Contribution analysis at 700 nm; (**b**) Contribution analysis at 900 nm; (**c**) Loss analysis at 700 nm; (**d**) Loss analysis at 900 nm.

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
