# Peer review of "Photon-Detection-Probability Simulation Method for CMOS Single-Photon Avalanche Diodes"

_sensors, 2020, doi:10.3390/s20020436_

Round 1
Reviewer 1 Report
Review report on the manuscript (Sensors-665422) “Photon-detection-probability simulation method for CMOS single-photon avalanche diodes” by Hsieh, Tsai, Tsui, Hsiao, and Lin
Hsieh et al. developed a fitting-free simulation procedure to predict PDP. The simulation results agree with the experimental data (in 800nm CMOS process) when the wavelength is larger than 600nm. Possible reasons for the discrepancy at shorter wavelengths were discussed. I have a few comments, and the authors should respond thoroughly before the manuscript can be acceptable.
Major comments:
What is the main reason to use Episil 800nm CMOS process, given that 350nm, 180nm or even more advanced CMOS technologies have been proven effective in manufacturing low-noise SPAD sensors? See Akita et al., “An imager using 2D single-photon avalanche diode array in 0.18-um CMOS for automotive LIDAR applications” Proc. Symp. VLSI Circuits, pp. £290-C291, (2017). Would the proposed method be applicable to other manufacturing processes?
The authors could adopt professional proof-reading service to proofread the English writing. Examples are not provided here, but all nouns should have proper articles and proper tenses should be used, for example.
Page 1, Line 25, “Due to their … and readiness for one-chip integration with CMOS circuits, SPADs have been …..”. This argument is misleading. SPADs (not necessarily CMOS) have been used in fluorescence lifetime imaging microscopy for a long time, although early systems required scanning. Were the authors meant to say that using SPAD arrays brings a step-change in the mentioned research areas? I believe the answer is yes, and I would suggest the authors also cite recent works in these areas. For example, (1). Homulle et al., “Compact solid-state CMOS single-photon detector array for in vivo NIR fluorescence lifetime oncology measurements,” Biomedical Optics Express, vol. 7, pp. 1797-1814. (2). Henderson et al., “A 192x128 time correlated SPAD image sensor in 40nm CMOS technology,” IEEE Solid-State Circuits, vol. 54, pp. 1907-1916 (2019).
When talking about the novelty of the proposed work, a diagram can be used to illustrate how different your approach is compared with existing models.
Try to make your English writing compact. For example replace ‘to extract the values of xx and yy (Line 172)’ with ‘to estimate xx and yy’. Again, please have professional proof-reading service to improve the writing.
Figure 6: The labels of the colour bars in Figs 6(a)-(c) are barely seen.
Figure 7: The dark count > 20kHz when the bias voltage > 29.1V. The transfer curve shows that DCR is highly sensitive to the bias voltage. What contributed to this behaviour?
Figure 9: Could the authors explain how the curves in Fig. 9(b) were normalised? And why the measured curves gave much better results than simulated results? So the proposed method is more robust when Vex is larger? Is the proposed method only suitable for long-wavelength applications?
Reviewer 2 Report
The paper is well written and provides a useful approach to evaluating PDP in SPADs without fitting parameters with best results at NIR wavelengths.
Page 2 line 62 it is stated that the SPAD has a diameter of 10 um but Fig. 1a shows a diameter of 10um. Please make consistent.
The size of the text in TCAD plots showing scales should be increased.
MATLAB code is mentioned on page 3. Could any code relevant to reproducing the results in the paper by others with access to Synopsys TCAD tools be provided (as an appendix or doi link)?
Fig. 7 The DCR increases linearly with Vex. Most authors measure an exponential increase. Is there a reason for this or should the y-axis be in log?
Fig. 8 Measured PDP vs wavelength show no Fabry-Perot reflections in the optical stack as is normally measured by others. Can you justify this?
Fig. 9b It is difficult to understand how this "normalisation" has been undertaken. Please give more details in the text or caption.
Page 7 The authors give a number of reasons for the discrepancy between measured and simulated results at 500nm. I am surprised that suitable parameters of defect density, minority carrier lifetime, diffusion length could not have been inferred and simulated to show that a closer match could be obtained. The conclusion is somewhat disappointing in this regard.
Page 2 BF2 -> BF3
Page 5 line 155-158 English: -> number of electrons are... number of holes are...
Page 8 Line 265 Pasb -> Pabs
Page 8 Line 267 are undertaking -> are being undertaken.
Fig. 10 Transimission -> Transmission
